# A novel computational approach for predicting complex phenotypes in *Drosophila* (starvation-sensitive and sterile) by deriving their gene expression signatures from public data

Dobril K. Ivanov[1,2]*, Gerrit Bostelmann[1], Benoit Lan-Leung[2], Julie Williams[2], Linda Partridge[3,4‡], Valentina Escott-Price[2‡], Janet M. Thornton[1‡]

**1** European Molecular Biology Laboratory, The European Bioinformatics Institute (EMBL-EBI), Hinxton, Cambridge, United Kingdom, **2** UK Dementia Research Institute at Cardiff (UKDRI), College of Biomedical and Life Sciences, Cardiff University, Cardiff, United Kingdom, **3** Max Planck Institute for Biology of Ageing, Cologne, Germany, **4** Institute of Healthy Ageing, and Department of Genetics, Evolution and Environment, UCL, London, United Kingdom

‡ These authors are joint senior authors on this work.
* IvanovD1@cardiff.ac.uk

## Abstract

Many research teams perform numerous genetic, transcriptomic, proteomic and other types of omic experiments to understand molecular, cellular and physiological mechanisms of disease and health. Often (but not always), the results of these experiments are deposited in publicly available repository databases. These data records often include phenotypic characteristics following genetic and environmental perturbations, with the aim of discovering underlying molecular mechanisms leading to the phenotypic responses. A constrained set of phenotypic characteristics is usually recorded and these are mostly hypothesis driven of possible to record within financial or practical constraints. We present a novel proof-of-principal computational approach for combining publicly available gene-expression data from control/mutant animal experiments that exhibit a particular phenotype, and we use this approach to predict unobserved phenotypic characteristics in new experiments (data derived from EBI's ArrayExpress and ExpressionAtlas respectively). We utilised available microarray gene-expression data for two phenotypes (starvation-sensitive and sterile) in *Drosophila*. The data were combined using a linear-mixed effects model with the inclusion of consecutive principal components to account for variability between experiments in conjunction with Gene Ontology enrichment analysis. We present how available data can be ranked in accordance to a phenotypic likelihood of exhibiting these two phenotypes using random forest. The results from our study show that it is possible to integrate seemingly different gene-expression microarray data and predict a potential phenotypic manifestation with a relatively high degree of confidence (>80% AUC). This provides thus far unexplored opportunities for inferring unknown and unbiased phenotypic characteristics from already performed experiments, in order to identify studies for future analyses. Molecular mechanisms associated with gene and environment perturbations are intrinsically linked and give

**Data Availability Statement:** All relevant data are within the manuscript and its Supporting Information files.

**Funding:** This work was supported by the UK Dementia Research Institute which receives its funding from DRI Ltd, funded by the UK Medical Research Council, Alzheimer's Society and Alzheimer's Research UK. The project was also part-funded by the European Regional Development Fund through the Welsh Government (CU-147; Dr Dobril Ivanov and Dr Benoit Lan-Leung) and by the Wellcome Trust Strategic Award (098565/Z/12/Z; Prof Linda Partridge).

**Competing interests:** The authors have declared that no competing interests exist.

rise to a variety of phenotypic manifestations. Therefore, unravelling the phenotypic spectrum can help to gain insights into disease mechanisms associated with gene and environmental perturbations. Our approach uses public data that are set to increase in volume, thus providing value for money.

## Introduction

Despite the flood of molecular omics data, with a few notable exceptions, such as the Genotype-Tissue Expression (GTEx) project [1], most datasets are rarely re-used, mainly due to challenges with combining the data from different sources. However, in most experimental studies, additional measures are made of biochemical, and physiological changes and of changes in the phenotypic characteristics that they bring about. Phenotypes can include, for instance, morphology, behaviour and pathology. Usually, a limited number of phenotypes are recorded, due to various study constraints. An intermediate phenotype, or sub-phenotype, is one that underlies the study phenotype, but crucially is influenced by fewer genes [2]. For instance, sub-phenotypes of Parkinson's Disease (PD) can include olfactory impairment, gut function disturbance, motor impairments and cognitive decline, each of which may be mediated by subsets of the genes that together result in PD pathology. Quantifying a wide variety of sub-phenotypes associated with animal models of a disease could therefore help to identify causal mechanisms.

The aim of the present study was to develop an *in-silico* approach for inferring unobserved phenotypic characteristics from published gene-expression data resulting from genetic or environmental perturbations. To do this, we generated molecular signatures for two target phenotypes in the fruit fly *Drosophila*, starvation stress response defective (starvation-sensitive) and sterile, using available gene-expression data. Using machine learning, we were able to show that these molecular signatures are able to reliably predict the starvation-sensitive and sterile phenotypic traits solely using expression datasets from studies where these phenotypes were not originally measured, thus adding value to already deposited data.

## Materials and methods

A schematic overview of the generation of a gene-expression molecular signature for a specific phenotype of interest is presented in Fig 1.

### Data collection

**Linking phenotypes to perturbed genes in *Drosophila*.**   In order to identify perturbed genes that lead to a particular phenotype, we downloaded several datasets from FlyBase (http://flybase.org/). These comprised: allele phenotypic data, synonyms, annotation identifiers, control vocabulary and alleles to gene identifiers. Using in-house custom programs, we parsed and linked all these identifiers with the phenotypic data. That is, for each FlyBase phenotype, we obtained a list of identifiers (e.g. FlyBase gene numbers, allele symbols, synonyms).

**Obtaining expression data from EBI's ArrayExpress.**   To maximise the number of experiments for each phenotype chosen for this study, we used the Affymetrix GeneChip Drosophila Genome 2.0 Array (EBI's ArrayExpress identifier A-AFFY-35). At the time of conducting the analysis, the largest number of experiments had been performed using the Affymetrix Genome 2.0 microarray platform (number of experiments: 330).

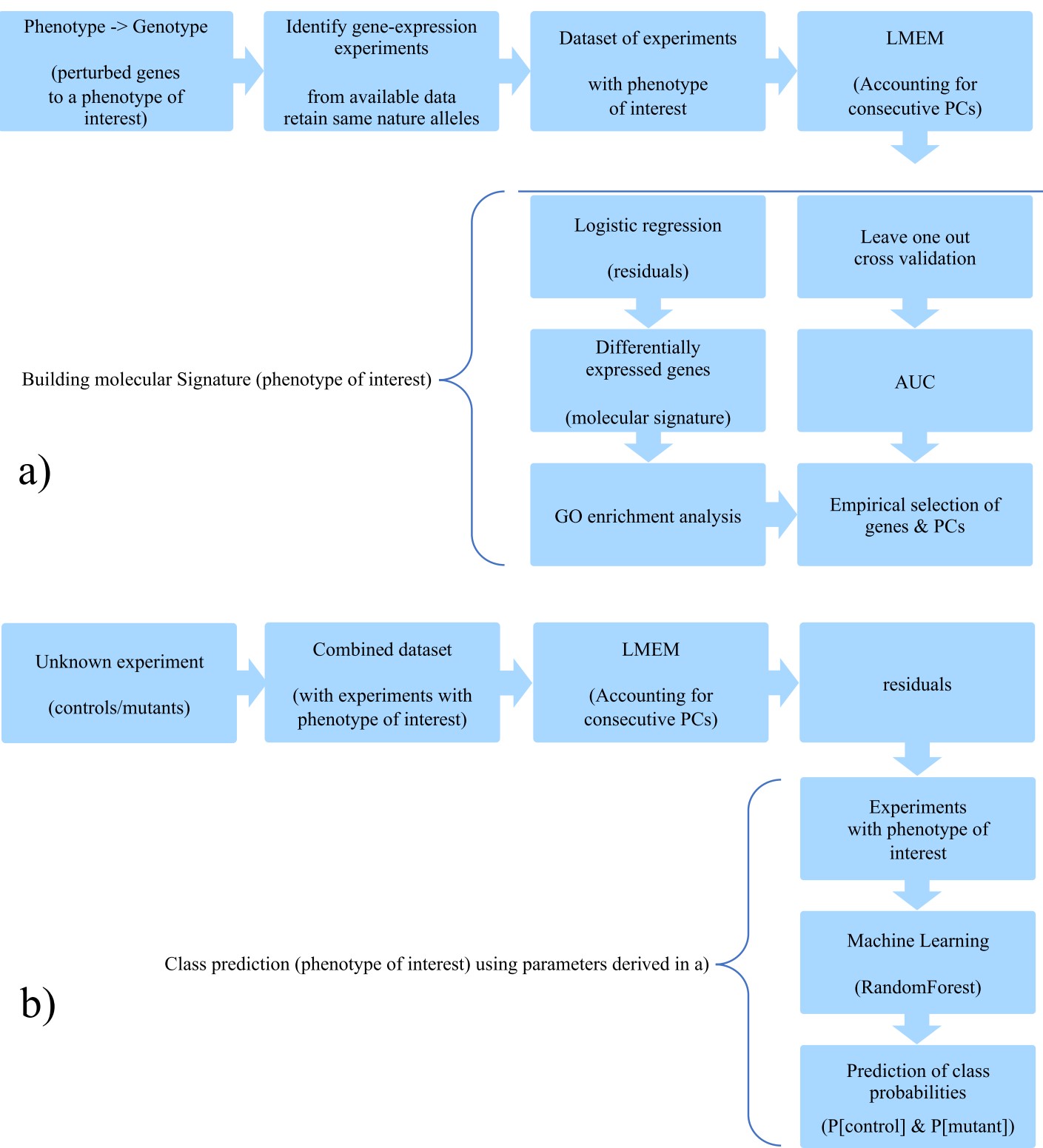

**Fig 1. Flow diagram of the overall generation of molecular signatures for a phenotype of interest.** a) Building the molecular signature and selecting model parameters for a particular phenotype. b) Predicting phenotypic manifestation in unknown experiments utilising the molecular signature.

Using the above-mentioned FlyBase identifiers (linking phenotypes to perturbed genes) we searched EBI's ArrayExpress for any potential match using the textual representation of EBI's web resource, i.e. REST-style queries. The identifiers were used as keywords to form a URL and the XML result was parsed using a custom-made Perl program. The nature of the allele constructs for experiments deposited in EBI's ArrayExpress does not follow a specific nomenclature and the authors/depositors are allowed relative freedom in describing the gene constructs. For example, EBI's ArrayExpress identifier E-GEOD-18576 lists a genotype description as a *DHR96* mutant. We did not assume that different allele constructs for the same gene will exhibit the same phenotype. Therefore, for each of the experiments that matched any of the FlyBase identifiers for the two target phenotypes, we manually curated the data first by reading all the accompanying manuscripts and subsequently retained experiments where the same allele construct was used. Furthermore, only experiments with raw gene-expression data (data with available raw cel files) were retained.

## Normalised gene-expression values

Raw gene-expression data (cel files) were downloaded from the EBI's ArrayExpress (https:// www.ebi.ac.uk/arrayexpress/). An 'experiment' throughout this manuscript was considered to be a set of control/mutant gene-expression microarray assays, submitted to EBI's ArrayExpress under the same identifier and exhibiting the phenotype of interest, unless otherwise specified (see Fig 2). Separately, for each experiment, the raw data were summarised and normalised by using the rma (bioconductor's package *affy* [3]). Log2-normalised expression data for all experiments that exhibited a particular phenotype were combined in a single dataset.

## Removal of batch effects within an experiment

Individual experiments for the two target phenotypes were examined for the presence of batch effects. For each ArrayExpress accession number, all individual microarray cel files were downloaded, including any microarray assays that did not exhibit the phenotypes in question but were submitted under the same ArrayExpress identifier. For each experiment, we performed principal component analysis (PCA) of the log2-normalised microarray expression data. Where significant batch effects were detected, we used bioconductor's *ber* package [4] to

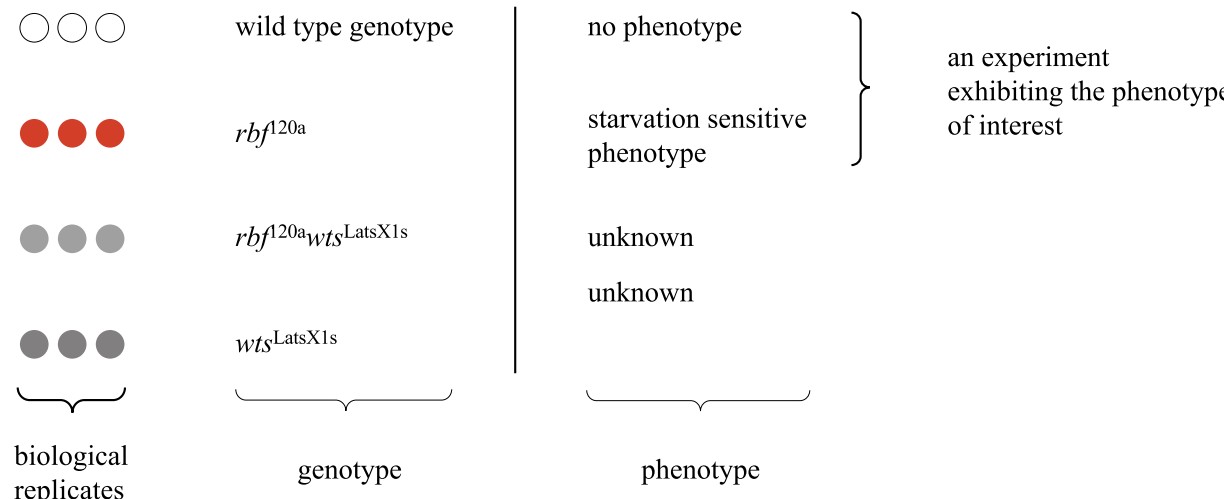

**Fig 2. Definition of an experiment exhibiting a phenotype of interest.** EBIs ArrayExpress identifier: E-GEOD-24978.

correct for them. For example, if an experiment that exhibited the phenotype of interest had sets of controls/mutants derived from different tissues, and that therefore exhibited significant heterogeneity in pattern of gene expression, the tissue effect was used as a factor in the batch effect correction.

## Generation of the molecular signatures (linear-mixed effects model)

A random intercept linear mixed-effects model (LMEM) was used to generate normalised residuals for each gene within the Affymetrix Genome 2.0 microarray, accounting for a number of consecutive principal components. Fixed and random effects comprised the principal components and the different experiments, respectively, with gene-expression as the dependent variable. The residuals were then used to perform a logistic regression to assess the statistical significance of each gene. For the LMEM, the *lmer* function in R was used. The number and nature of the underlying biological and technical factors that differ between the different experiments are largely unknown. In order to determine how many principal components to use, the molecular signatures for the two target phenotypes were generated using LMEM, including a number of consecutive principal components to account for these biological/technical effects, e.g. sex, tissue. The consecutive principal components used started with using LMEM with no principal components progressing up to a LMEM with the first 7 consecutive principal components included (8 different models).

## Gene Ontology (GO) enrichment analysis

The Wilcoxon rank sum test, as implemented in Catmap [5], was used to perform functional analysis to test for significant enrichment of Gene Ontology categories. Ranks of genes were based on the p-value derived from the logistic regression, irrespective of beta-coefficients. To account for multiple hypotheses testing the Benjamini-Hochberg false discovery rate was used (FDR). To assess if there was a significant enrichment of GO terms associated with the two target phenotypes of interest in the derived molecular signatures, we selected GO terms that we considered representative of the two phenotypes (S1 and S2 Figs in S1 File).

## Leave-one-out cross-validation

To assess how well the molecular signatures could be used to predict the target phenotype in other experiments that exhibit a phenotype of interest, we used *randomForest* package in R (default parameters with 1,000 trees). We used a leave-one-out cross-validation (LOOCV) in order to calculate an area under the curve (AUC). Iteratively for all experiments we left one experiment out and derived the molecular signature using the rest of the experiments that exhibited the target phenotype. For example, one iteration comprised removing the controls/mutants, part of the *crol* experiment (starvation-sensitive) and generating the molecular signature using the rest of the experiments (*dhr96*, *mir14*, *p53* and *rbf*). Crucially, we derived the residuals from the random intercept LMEM, along with consecutive principal components, for all experiments that exhibited the target phenotype, and then left one experiment out. This ensured that the model was corrected for underlying technical factors before performing the LOOCV. The AUC was calculated using the class (control/mutant) probabilities derived from the *randomForest* package, using the top 200 genes from the molecular signature (based on the p-values from the logistic regression). We also tested a different number of top genes (from 50 to 3,000 genes, S6 and S7 Figs in S1 File for the starvation-sensitive and sterile phenotypes respectively). In addition, we also formally tested if the mean of the class probabilities was different from 0.5 using a t-test, separately for controls and mutants, for the left-one-out

experiment. The probability of 0.5 is the null hypothesis and it is equivalent to a random assignment of the controls/mutants.

## Predicting the presence of phenotypic expression in freely available data

Similarly to the LOOCV, we used the molecular signature (top 200 genes based on the p-value from the logistic regression) for the starvation-sensitive and sterile phenotypes to predict the presence of the phenotypes in all available data in ArrayAtlas (Affymetrix GeneChip Drosophila Genome 2.0 Array). Iteratively for each deposited experiment in ArrayAtlas, we first derived residuals from a random intercept LMEM, including consecutive PCs, from the combined log2-normalised data for the experiment and the experiments that were part of the two phenotypes (starvation-sensitive and sterile). This ensured that we accounted for any technical variability between experiments. These residuals were then used to derive the probabilities for class (control/mutant) separation with the *randomForest* package in R. Each individual control/mutant sample within an experiment was assigned a class probability (control or mutant). For each class (control or mutant) the probabilities were averaged across the number of samples, separately for controls and mutants. This mean probability was used to infer quantitatively the target phenotype.

## Results

### Experiments and expression data

Using the above protocol, we identified five and six experiments, respectively, with specific perturbed genes for which gene-expression data for the starvation stress response defective (FlyBase control vocabulary identifier FBcv:0000708) and the female sterile (FBcv:0000366) target phenotypes were available. These were *dhr96* (E-GEOD-18576), *mir-14* (E-GEOD-20202), *rbf* (E-GEOD-38430), *p53* (E-GEOD-37404) and *crol* (E-GEOD-8775) for the starvation sensitive phenotype and *loj* (E-GEOD-10940), *ovo* (E-GEOD-48145), *pxt* (E-GEOD-29815), *su(HW)* (E-GEOD-36528), *ttk* (E-GEOD-42758) and *vret* (E-GEOD-30360) for the sterile phenotype. Additional information can be found in S1 and S2 Tables in S1 File. Following normalisation and excluding transcripts that did not match any known or predicted gene, there were 12,630 genes left for analysis. The normalised gene-expression data are available upon request.

### GO-terms enrichment analysis

Figs 3 and 4 show the results for the GO-terms associated with the two target phenotypes respectively (full numerical data are shown in S5 and S6 Tables in S1 File). Enrichment of starvation-related GO terms for the starvation-sensitive phenotype was observed for LMEM with the inclusion of one to four PCs (Fig 3). In contrast, sterile-related GO terms were found to be mostly enriched with LMEM without the inclusion of PCs (Fig 4). This suggests that there is more inter-experiment variability associated with the starvation-sensitive phenotype as compared to the sterile. All of the individual gene perturbation experiments that exhibited the sterile phenotype comprised female flies and more homogeneous tissue used to derive the expression data (S2 Table in S1 File), whereas the individual experiments for the starvation-sensitive phenotype were mixed sex and the expression data were derived from a variety of tissues (S1 Table in S1 File).

We also performed a GO enrichment analysis associated with individual control/mutant experiments exhibiting the two target phenotypes (e.g. *crol* part of E-GEOD-8775). Ranks of genes were derived using the *limma* package in R. Only two experiments showed any

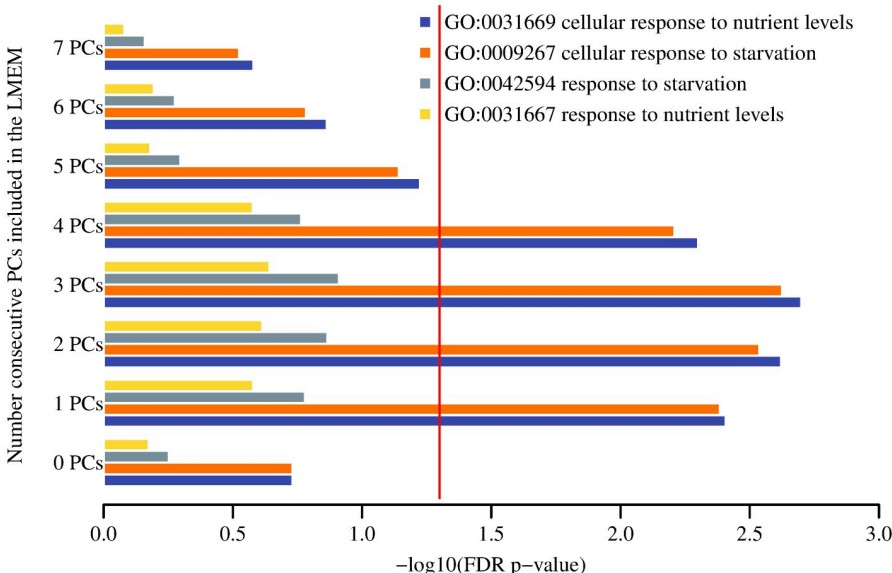

**Fig 3. Top GO terms for the starvation-sensitive molecular signature.** Red vertical line represents FDR p-value 0.05.

statistically significant evidence of GO-terms enrichment associated with the starvation pheno-type (*crol* and *p53*; S3 Table in S1 File), whereas all of the experiments that were identified to exhibit the sterile phenotype showed statistically significant enrichment of reproduction-related GO terms (S4 Table in S1 File).

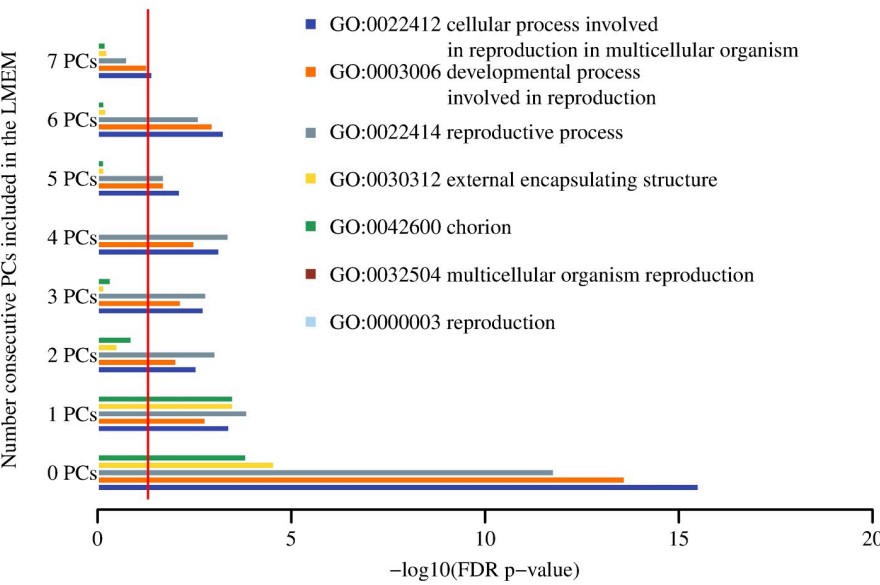

**Fig 4. Top GO terms for the sterile molecular signature.** Red vertical line represents FDR p-value 0.05.

## Removal of batch effects within an experiment

Only one experiment (*loj*), with the sterile phenotype, exhibited a significant batch effect. The controls and mutants comprised two tissues (abdomen and head/thorax). We used the *ber* package to correct for the batch effect using the tissue as a factor. We observed two clusters for the first PC (89.34% variance explained) that separated the *loj* by tissue (S3a Fig in S1 File). Correcting for the tissue batch effect eliminated the tissue separation and the *loj* controls/ mutants separated by the second PC (S3b Fig in S1 File).

## Determining the number of PCs for unwanted variation

The maximum AUC for the leave-one-out cross-validation for the starvation sensitive phenotype was 97% with six consecutive PCs and 85% with LMEM with no PCs for the sterile phenotype (Figs 5 and 6).

Nevertheless, GO term enrichment analysis showed that the statistical significance of starvation-related GO terms disappeared (FDR p-value >0.05) when the first five or six PCs were included in the LMEM (Fig 3). GO terms enrichment results for the sterile phenotype are shown in Fig 4. Furthermore, PCA of the residuals of the starvation sensitive LMEM with five or six PCs showed near complete separation of the controls and mutants (S4f and S4g Fig in S1 File). Taken together, these results suggest that the first four PCs account for biological/technical variability, that the overall molecular signature is enriched with starvation-related GO terms, and the 5th and 6th PCs account for the starvation-sensitive phenotype. We hypothesise that when we account for the first 5–6 PCs, the signal that is left is a form of global gene-expression regulation following a gene perturbation. Thus, accounting for the first five or six PCs results in a prediction of the class separation, rather than the manifestation of the phenotype. A gene perturbation disrupts the global gene-expression equilibrium and results in differential expression of compensatory gene mechanisms. In other words, control/mutant experiments with seemingly different gene perturbations may result in a higher than expected by chance overlap of differentially expressed genes, i.e. genes that are part of the compensatory gene-expression regulatory network. In order to test this hypothesis, we performed 1,000 permutations, whereby we chose five random control/mutant experiments from EBI's ArrayExpress. The number of controls/mutants per experiment was matched to the number of controls/mutants in the five experiments for the starvation-sensitive phenotype. Thus, the number of controls/mutants in a randomly chosen experiment was reduced to match the number of controls/mutants in S1 Table in S1 File. For each of these experiments we derived normalised gene-expression values using the same procedure as for the starvation-sensitive phenotype. We derived differentially expressed genes using the *limma* package in R. For each of these random sets of experiments, we selected the top 200 genes and calculated the number of genes that overlap within each set of experiments in a pairwise manner. For each of these permutations we calculated the median of the -log10 of the p-value for each pairwise overlap using hypergeometric distribution. We compared these results to the pairwise overlap of random 200 genes as part of 1,000 sets of experiments. The distributions of the results for the random 1,000 sets of experiments and for what is expected by chance are shown in Fig 7.

The results presented in Fig 7 clearly show that a random combination of sets of five experiments exhibit a significantly greater number of differentially expressed genes that overlap between the experiments as compared to purely by chance alone. This observation has been also reported in humans [6]. Thus, for the leave-one-out cross-validation for the starvation-sensitive phenotype we used the first four PCs to account for biological/technical variation. For the sterile phenotype we did not use PCs (LMEM with 0 PCs). PCA graphs for the sterile molecular signature LMEM with 0 to 7 PCs are shown in S5 Fig in S1 File. For the calculation

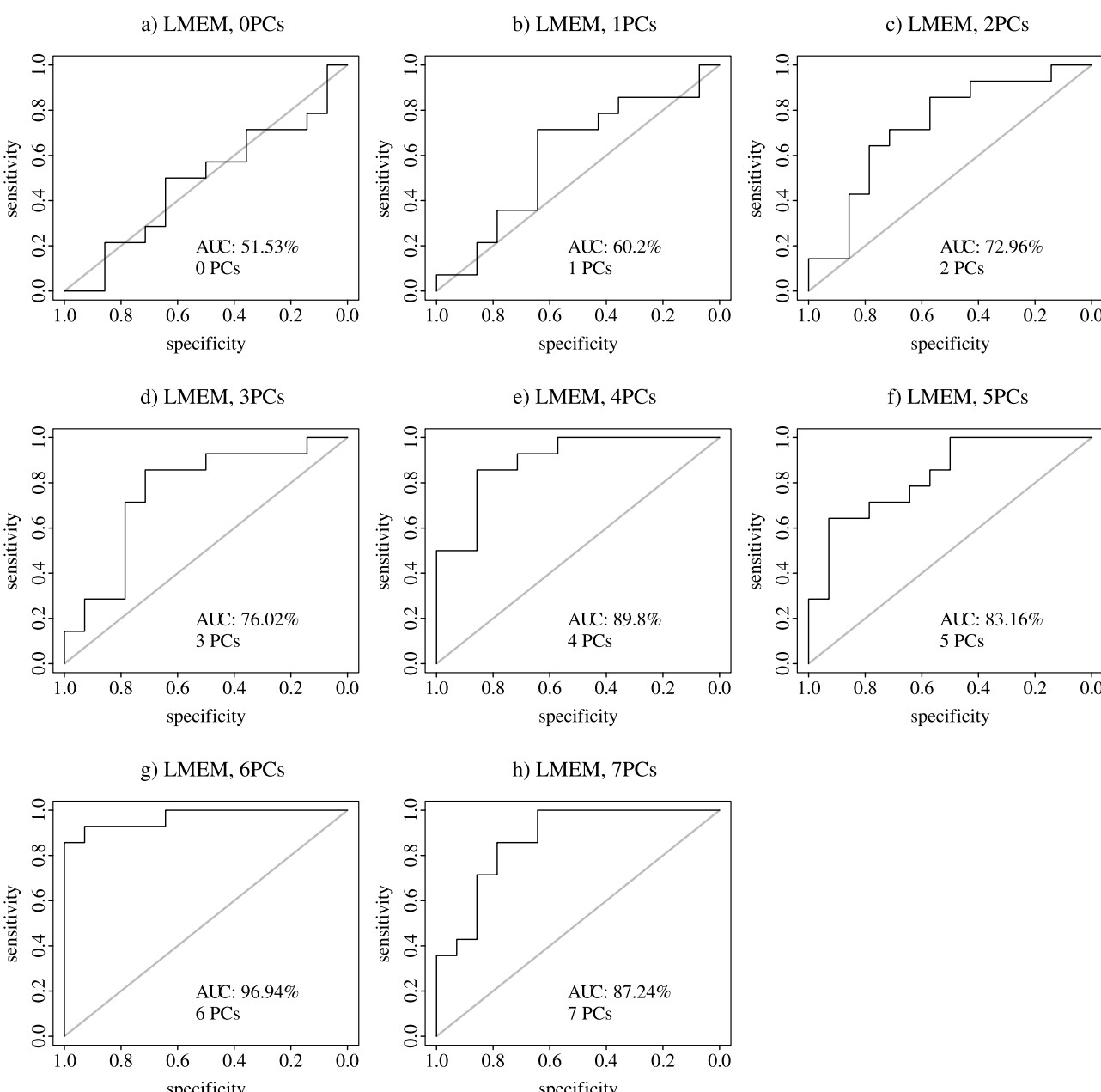

**Fig 5. Starvation-sensitive phenotype, leave-one-out cross-validation AUC.** AUC- Area Under the Curve; a through h LMEM with 0 to 7 PCs.

of the AUC for the LOOCV we tested a range of top genes (from 50 to 3,000). For the starvation-sensitive phenotype there was not a difference in the AUC with different number of top genes, although choosing more genes resulted in a slightly higher AUC (50 genes 87.76% AUC; 3,000 genes 90.31% AUC; S6 Fig in S1 File with 4PCs). The opposite was noted with the sterile phenotype, fewer number of top genes resulted in higher AUC (50 genes 90.58% AUC; 3,000 genes 73.68% AUC; S7 Fig in S1 File with 0PCs). These trends could potentially reflect the size of the transcriptional network involved in both phenotype, for example it has been

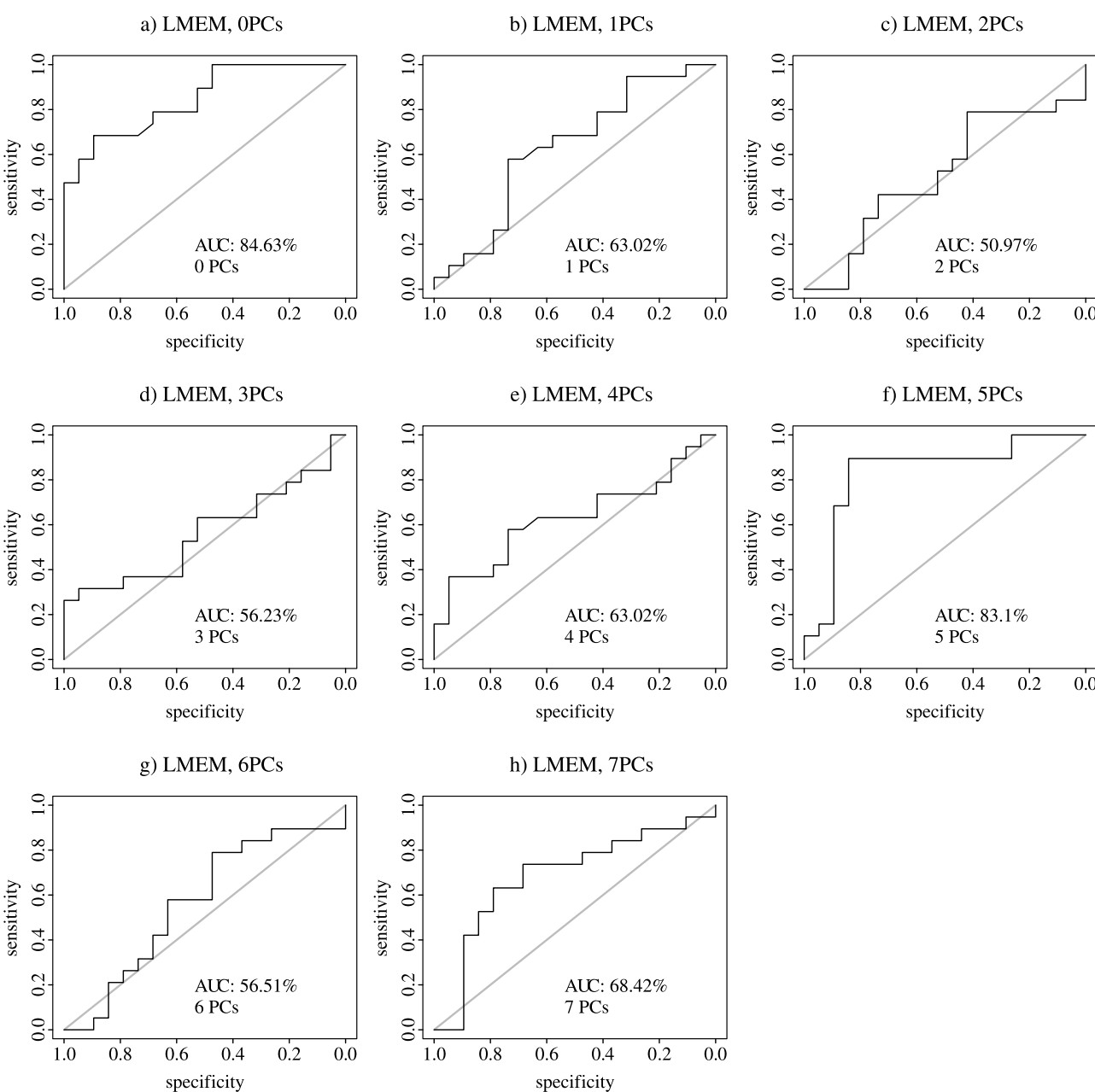

**Fig 6. Sterile phenotype, leave-one-out cross-validation AUC.** AUC- Area Under the Curve; a through h LMEM with 0 to 7 PCs.

previously reported that the starvation stress resistance involves transcriptional response of ~25% of the genome in *Drosophila* [7].

The mean distribution of the control/mutant class probabilities from the random forest for both the starvation-sensitive and sterile phenotypes were significantly different from 0.5 (Table 1). The results in Table 1, along with the AUC for both phenotypes (Figs 5 and 6), show that we can confidently predict the phenotypic manifestation of a separate experiment that exhibits the phenotype of interest.

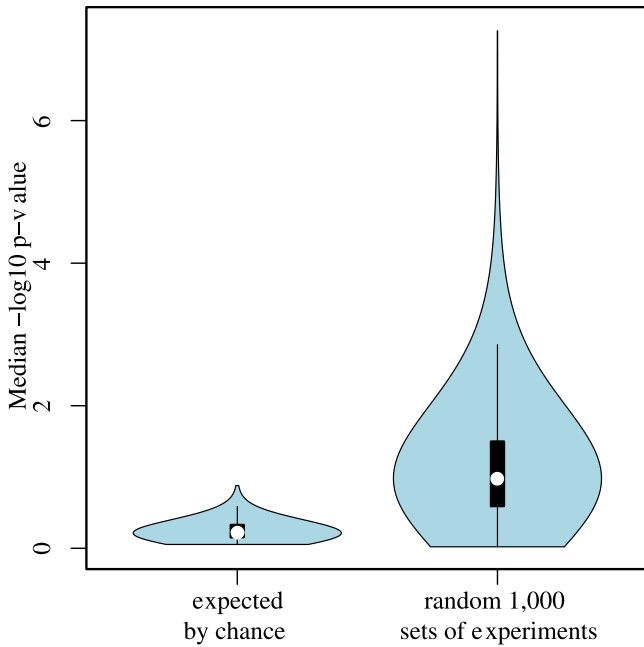

**Fig 7. Distribution of the pairwise overlap of genes in 1,000 random sets of five experiments, derived from ArrayExpress, as compared to expected by chance.** Y-axis- Median -$\log_{10}$ hypergeometric p-value for significance of pairwise overlap.

## Predicting freely available experiments for the presence of both phenotypes

In order to obtain freely available experiments we utilised EBI's ExpressionAtlas (https://www.ebi.ac.uk/gxa/home) instead of ArrayExpress. We used EBI's ExpressionAtlas due to the availability of normalised gene-expression values for a large number of the already available raw cel gene-expression data in ArrayExpress. This eliminated the need to normalise all of the available raw gene-expression data within ArrayExpress. For all experiments available in EBI's ExpressionAtlas (total number of control/mutant experiments at the time of conducting the study: 211) we used the molecular signatures for the starvation sensitive and sterile phenotypes to derive a mean probability separately for controls and mutants in an experiment. The mean mutant probability was used to suggest a degree of phenotypic manifestation. Ranking of all available experiments is given in S7 and S8 Tables in S1 File for the starvation-sensitive and sterile phenotypes respectively.

**Ranking EBI's ExpressionAtlas experiments for the starvation-sensitive phenotype.** The top three ranked experiments were all already used to generate the molecular signature (*dhr96*, *crol* and *rbf*), thus it is not unexpected that we can predict these experiments with the

**Table 1. One sample t-test for class probabilities (controls/mutants) in the two phenotypes following LOOCV.**

| Class | Phenotype | |
|---|---|---|
| | starvation-sensitive p-value (μ = 0.5) | Sterile p-value (μ = 0.5) |
| Controls | $5.72 \times 10^{-03}$ | $3.87 \times 10^{-03}$ |
| Mutants | $5.84 \times 10^{-03}$ | $3.10 \times 10^{-03}$ |

highest accuracy. The *p53* (E-GEOD-37404) and *mir-14* (E-GEOD-20202) experiments are not included in the EBI's ExpressionAtlas datasets.

For the rest of the freely available experiments available in EBI's ExpressionAtlas we found no results from a direct lab-based assay of the starvation sensitivity. Nevertheless, for some of the top-ranked experiments we found additional evidence that can be potentially used to support the results from our prediction. All three gene mutants (*rbf*$^{120a}$, *rbf*$^{120a}$ *wts*$^{latsX1}$ and *wts*$^{latsX1}$), part of an experiment (E-GEOD-24978) were ranked with mutant class probabilities of 83%, 74% and 64% respectively. The two genes, *rbf* and *wts* regulate cell proliferation via the p16 and Hippo tumour suppressor pathways. There is only a direct lab-based measurement of the starvation-sensitive phenotype of *rbf*$^{120a}$, which was used as part of the molecular signature. We speculate that the *wts*$^{latsX1}$ and the double-mutant *rbf*$^{120a}$ *wts*$^{latsX1}$ may also exhibit starvation-sensitive phenotype.

Several of the top-ranked experiments included fly lines from the *Drosophila* Genetic Reference Panel (DGRP) [8]. These included genes (*esg*, *Pdcd4*, *mub*, *Gbs-70E*) that were reported to exhibit a reduced starvation resistance, tested at six weeks.

**Ranking EBI's ExpressionAtlas experiments for sterile phenotype.** The top four ranked experiments in the EBI's ExpressionAtlas comprise four already used control/mutant experiments for the sterile molecular signature (*ovo* (*ovo* and *ovo/cako*) and *loj* (head and thorax)), thus it is not surprising that we can detect these with high accuracy. The rest of the experiments, part of the molecular signature, were not analysed as part of EBI's ExpressionAtlas (not all experiments from ArrayExpress are analysed in ExpressionAtlas). Similarly to the starvation-sensitive molecular signature we found no direct evidence that the top-ranked experiments will exhibit the sterile phenotype. Nevertheless, there was additional evidence for some of the top-ranked experiments. For example, experiment E-GEOD-55187 comprises *sesb*$^1$ homozygous female mutants that are predicted to exhibit the sterile phenotype with mean probability of 85% across the individual mutants. *Sesb*$^1$ is listed as female sterile in flybase (http://flybase.org/reports/FBal0015434-phenotypic_data_sub). Due to lack of information, we could not verify whether the gene-mutant shown as sterile [9] is exactly the same as the gene-mutants with the microarray data in EBI's ArrayExpress [10]. Similarly, in experiment E-MTAB-3546 [11], 3-week reproductive diapause under cold conditions (11C) was predicted to exhibit the sterile phenotype with a mean mutant probability of 91% across the individual mutants. Clearly, the mutant female flies are very likely to exhibit the sterile phenotype as they were induced into a diapause that is associated with a reproductive arrest. The 10 and 40 days aged dietary restricted female flies (E-GEOD-26726) also showed evidence of the sterile phenotype (84% and 79% respectively). There is a well-defined reduction in daily and lifetime fecundity under dietary restriction [12], therefore it is more than likely that the 10 and 40 days old flies will exhibit the sterile phenotype.

## Discussion

In this paper we present a novel computational approach for integrating gene-expression data for two specific phenotypes (starvation-sensitive and sterile) in *Drosophila* from the vast and largely unutilised freely available public repositories. This integration is multi-layered with phenotypic information derived from a species-specific database (FlyBase) and gene-expression from the largest repository of publicly available genomic data, the ExpressionAtlas at the European Bioinformatics Institute. Crucially, we present an approach to utilise gene-expression data generated by completely independent groups across the scientific community.

The results of this proof-of-concept study show that it is possible to integrate seemingly different gene-expression microarray data using a combination of linear-mixed effect models and

principal components analyses and predict a potential phenotypic manifestation with a relatively high degree of confidence. Nevertheless, the applicability of this methodology to capture a wide range of phenotypes and organisms requires a considerable amount of additional work that is beyond the scope of this article.

The premise of our methodology is based upon the assumption that specific cellular and physiological phenotypes are underlined by or associated with similar gene-expression changes. In addition, the number of such gene-expression changes that are shared between different perturbations and are associated with a specific phenotype, is likely to differ between different phenotypes. Currently, there is no simple way to derive a set number of gene-expression changes that describe a particular phenotype and this number is also likely to depend on the nature of the phenotype. We used an empirically derived number of genes for the two phenotypes that we tested (top 200 genes, based on p-value for differential expression), although this selection can potentially be automated using a different number of genes. Our approach might not be directly applicable if a specific phenotype is underlined by independent biological pathways or caused by mechanisms that do not result in changes in gene-expression. Nevertheless, additional genomic measurements can be incorporated as and when they become available. Furthermore, our methodology relies on freely available gene-expression data, which is only set to increase [13]. Thus, with the increase in repository data, our approach has a great potential to estimate relative degree of independence of biological pathways that influence or give rise to specific phenotypes.

Biological phenotypes are rarely binary features, although they often get binarised for ease of use, for example gravitaxis defective phenotype (movement away from the source of gravity) can be expressed as defective/normal or a more complex measure can be used to account for the continuous nature of the phenotype [14]. Nevertheless, even with considerable efforts to standardise experimental protocols and measurement assays, differences will be exhibited between laboratories across the world. As such, it is difficult to utilise the continuous phenotype response measurements. In this study we only considered control/mutant type experiments. For such experiments the measured phenotypes can be taken as relative with respect to controls, thus minimising the differences in protocols. Nevertheless, for most such experiments in *Drosophila*, there is no unified system/database that collects and archives the outcomes of such measurements and currently these have to be extracted manually from the corresponding manuscripts and assessment made on how similar the protocols are. Our methodology of predicting potential phenotypic manifestation uses a machine learning approach, that is random forest. This could potentially be used to infer the two phenotypes probabilistically, although it is unclear what the relationship is between the similarity in gene-expression and the degree of phenotype manifestation.

Although our study utilises gene-expression microarray data and such type of data is clearly superseded by RNA sequencing [13], we do not foresee any major challenges in adopting our methodology to work with RNA-seq data. For example, raw RNA-seq counts can be relatively easily transformed into transcripts per million (TPM) and log2 of TPM can be used in the linear-mixed effect models.

Our methodology relies on linear-mixed effect models accounting for unwanted biological effects in the form of principal components. In order to estimate the number of PCs we utilised Gene Ontology enrichment analysis, whereby we chose consecutive number of PCs to maximise GO enrichment. One of the potential limitations is that there might be some degree of circularity when using GO terms to define phenotypic enrichment, since GO categories could have been partially defined using similar data. The other limitation is that the combination of PCs and linear-mixed effect model is likely to be overconservative, such that some variation in the phenotype of interest maybe already included in the PCs. Other approaches, such as

probabilistic estimation of expression residuals (PEER) [15] could be used to facilitate estimation of unwanted factors.

The proof-of-concept study presented here is a novel approach of predicting the manifestation of two phenotypes in *Drosophila* from gene-expression data. While, similar attempts have been previously performed [16–19], these studies rely on a single or a few well-defined datasets with few measured phenotypes. Our approach goes beyond single studies and it is not restricted to selective phenotypic measurements in a few datasets. The methodology described here captures the diverse genetic background and gene-perturbations from all the publicly available repository data and links them to phenotypic characteristics, thereby adding value to already deposited and largely unutilised data.

## Supporting information

**S1 File.**
(PDF)

## Acknowledgments

We would like to thank the Advanced Research Computing at Cardiff (ARCCA) and EMBL-EBI for providing computational resources.

## Author Contributions

**Conceptualization:** Dobril K. Ivanov, Julie Williams, Linda Partridge, Valentina Escott-Price, Janet M. Thornton.

**Data curation:** Dobril K. Ivanov, Gerrit Bostelmann, Benoit Lan-Leung.

**Formal analysis:** Dobril K. Ivanov, Gerrit Bostelmann, Benoit Lan-Leung.

**Investigation:** Dobril K. Ivanov, Gerrit Bostelmann, Benoit Lan-Leung.

**Methodology:** Dobril K. Ivanov, Gerrit Bostelmann, Benoit Lan-Leung, Valentina Escott-Price, Janet M. Thornton.

**Software:** Dobril K. Ivanov, Gerrit Bostelmann, Benoit Lan-Leung, Valentina Escott-Price.

**Supervision:** Dobril K. Ivanov, Julie Williams, Linda Partridge, Valentina Escott-Price, Janet M. Thornton.

**Visualization:** Dobril K. Ivanov, Gerrit Bostelmann, Benoit Lan-Leung, Linda Partridge, Valentina Escott-Price, Janet M. Thornton.

**Writing – original draft:** Dobril K. Ivanov, Valentina Escott-Price, Janet M. Thornton.

**Writing – review & editing:** Dobril K. Ivanov, Gerrit Bostelmann, Benoit Lan-Leung, Julie Williams, Linda Partridge, Valentina Escott-Price, Janet M. Thornton.

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
