## [Decision Letter · Decision Letter 0]

9 Jul 2020

PONE-D-20-03484

A novel computational approach for predicting complex phenotypes by deriving their gene expression signatures from public data

PLOS ONE

Dear Dr. Ivanov,

Thank you for submitting your manuscript to PLOS ONE. After careful consideration, we feel that it has merit but does not fully meet PLOS ONE’s publication criteria as it currently stands. Therefore, we invite you to submit a revised version of the manuscript that addresses the points raised during the review process.

We look forward to receiving your revised manuscript.

Kind regards,

Xia Li, Ph.D.

Academic Editor

PLOS ONE

Journal Requirements:

2.Thank you for stating the following in the Acknowledgments Section of your manuscript:

[This work was supported by the

UK Dementia Research Institute which receives its funding from DRI Ltd, funded by

the UK Medical Research Council, Alzheimer's Society and Alzheimer's Research

UK. The project was also part-funded by the European Regional Development Fund

through the Welsh Government.]

 [The author(s) received no specific funding for this work.]

Additional Editor Comments (if provided):

According to the reviewers' comments, my final decision is "Major" revision.

Reviewers' comments:

Reviewer's Responses to Questions

**Comments to the Author**

1. Is the manuscript technically sound, and do the data support the conclusions?

Reviewer #1: Yes

Reviewer #2: Yes

Reviewer #3: Yes

2. Has the statistical analysis been performed appropriately and rigorously? 

Reviewer #1: Yes

Reviewer #2: Yes

Reviewer #3: Yes

3. Have the authors made all data underlying the findings in their manuscript fully available?

Reviewer #1: Yes

Reviewer #2: Yes

Reviewer #3: Yes

4. Is the manuscript presented in an intelligible fashion and written in standard English?

Reviewer #1: Yes

Reviewer #2: Yes

Reviewer #3: Yes

5. Review Comments to the Author

Reviewer #1: The manuscript is written well and is understandable. However I think it should be made clearer in the abstract and within the text that both Array Express and Expression Atlas were used and an explanation of why the Expression Atlas was used.

Reviewer #2: Since the data was restricted to starvation and sterility, would suggest that the title reflect so, as phenotype is a very big bracket. And it is likely that the spectrum would vary between phenotypes e.g. colour, height etc vs binary cases like sterility. Also the data encompasses only Dropsophilia, so suggestion to be safer to be narrow in coverage in title and rest of the writing. By all means discuss on applicability across multiple organisms and phenotype in discussion bearing in mind limitations. It would be good to discuss applicability of towards different ranges of phenotypes. There is also a claim of capturing environmental influences, which is not really reflected in the results. Would suggest limiting claims and overall applicability, otherwise a lot of work should be done to substantiate those claims - multiple organisms, multiple range of phenotypes, etc.

Reviewer #3: The paper by Dobril K. Ivanov et al. demonstrated a combination of linear mixed effect models and principal components analyses approach for integrating gene expression data for specific phenotypes from independent labs widely available in public repositories. As a proof-of-concept study to show the promising in-silico approach for inferring phenotypic characteristics behind the published gene-expression data resulting from genetic or environmental perturbations, the data they selected is excellent for testing their methods and hypothesis. The results they presented are partial but convincing. I would like to see this approach be further explored, tested, and improved by more computational researchers. Along this line, since there are still some ambiguous or arbitrary steps or parameters in the study, I recommend that this paper only be accepted after the following major comments/issues are resolved:

The two represented phenotypes(starvation sensitive and sterile) were nicely investigated and presented. But in the current workflow, there are few steps that involve manual curation, like the selection of expression data and the representative GO terms. To show the application's universality, please add at least one additional test phenotype of interest that the related experiments and significant enrichment of GO terms were selected by predefined rules or algorithms.

The current flow diagram in Fig.1 is quite confusing, at least to me. Please consider separating the signature finding and phenotype prediction into two parts. Meanwhile, since the paper already discussed the complexities of the relation between the gene-expression changes and the phenotypes and described the limitation of the current approach. If possible, I would like to see an evaluation part by the end of the signature-finding section to evaluate whether the selected phenotype could be significantly presented or identified by the expression profiles of the currently available microarray experiments.

In the cross-validation section, the paper shows "the AUC was calculated using the class (control/mutant) probabilities derived from the randomForest package, using the top 200 genes from the molecular signature (based on the p-values from the logistic regression)". How this 200 was determined, whether this top-N influence the AUCs? I would like to know the results of a series of Ns to have a better understanding of this step. And prefer a more rigorous or data-driven way to calculate or determine the numbers of the signature genes for different phenotypes.

6. PLOS authors have the option to publish the peer review history of their article (what does this mean?). If published, this will include your full peer review and any attached files.

Reviewer #1: No

Reviewer #2: No

Reviewer #3: No

---

## [Author Response · Author response to Decision Letter 0]

14 Aug 2020

Reponse to Reviewers

RE: PONE-D-20-03484R1

"A novel computational approach for predicting complex phenotypes by deriving their gene expression signatures from public data" by Ivanov et al.

We would like to express our gratitude to the reviewers for their time and constructive comments of our manuscript, especially during these difficult times.

We have amended the main text of the manuscript and the supplementary materials taking into account and addressing all of the reviewer's comments. We believe that this has improved the manuscript.

We believe that the detailed response and major revision of the manuscript will be satisfactory to the reviewers and the Editor and the manuscript will be accepted for a publication in PLOS ONE.

Reviewer #1:

The manuscript is written well and is understandable. However I think it should be made clearer in the abstract and within the text that both Array Express and Expression Atlas were used and an explanation of why the Expression Atlas was used.

Response:

We are pleased to see that the Reviewer thought that the manuscript was well written and understandable. We have amended the abstract to include that publicly available gene-expression and new experiments data were derived from EBI's Array Express and Expression Atlas respectively. We have also provided an explanation why the Expression Atlas was used. We utilised EBI's Expression Atlas due to the availability of normalised gene-expression values and contrasts for individual experiments. Without this, we would have needed to normalise all the raw microarray cel data files within EBI's Array Express. This would have required a substantial amount of time and resources. The text (section "Predicting freely available experiments for the presence of both phenotypes") has been amended to include an explanation of why we used the normalised data within Expression Atlas and not the raw data in Array Express. We think that adding the explanation above makes the applied methodology clearer.

Reviewer #2:

Since the data was restricted to starvation and sterility, would suggest that the title reflect so, as phenotype is a very big bracket. And it is likely that the spectrum would vary between phenotypes e.g. colour, height etc vs binary cases like sterility. Also the data encompasses only Dropsophilia, so suggestion to be safer to be narrow in coverage in title and rest of the writing. By all means discuss on applicability across multiple organisms and phenotype in discussion bearing in mind limitations. It would be good to discuss applicability of towards different ranges of phenotypes. There is also a claim of capturing environmental influences, which is not really reflected in the results. Would suggest limiting claims and overall applicability, otherwise a lot of work should be done to substantiate those claims - multiple organisms, multiple range of phenotypes, etc.

Response:

We thank the reviewer for the constructive criticism and helpful suggestions. Yes, the suggestion is very helpful and we have amended the title along with the manuscript body to reflect the case use of two phenotypes and that it was only tested in Drosophila.

The new title is: "A novel computational approach for predicting complex phenotypes in Drosophila (starvation-sensitive and sterile) by deriving their gene expression signatures from public data".

The Abstract and Introduction were amended throughout to be more specific that we have generated gene-expression signatures for two phenotypes in Drosophila. We completely agree that it is safer to narrow the applicability. We have also amended the Discussion to be more precise in the range of claims. To this effect, the Discussion was amended to always specify that the results and claims refer to the two specific Drosophila phenotypes. We also included a new paragraph in the Discussion section regarding different ranges of phenotypes and more specifically binary vs. continuous measures of phenotypes.

Throughout the text we used the term environmental perturbation in a very limited scope. We do not claim that the derived gene-expression results for the two phenotypes capture environmental influences. That is, we were interested if genetic (e.g. gene knock-out) or environmental perturbations can induce or result in the presence of the two tested phenotypes. The term environmental perturbation was meant to mean for example a chemical compound or another type of intervention that has induced the phenotypes under investigation. For example, an ArrayAtlas experiment (E-MTAB-3546; 3-week reproductive diapause under cold conditions (11C)) was predicted to exhibit the sterile phenotype with a mean mutant probability of 91% across the individual mutants within that experiment. The cold conditions are an environmental perturbation that induces the reproductive diapause. In this specific instance we are able to confidently predict the phenotypic manifestation.

Reviewer #3:

The paper by Dobril K. Ivanov et al. demonstrated a combination of linear mixed effect models and principal components analyses approach for integrating gene expression data for specific phenotypes from independent labs widely available in public repositories. As a proof-of-concept study to show the promising in-silico approach for inferring phenotypic characteristics behind the published gene-expression data resulting from genetic or environmental perturbations, the data they selected is excellent for testing their methods and hypothesis. The results they presented are partial but convincing. I would like to see this approach be further explored, tested, and improved by more computational researchers. Along this line, since there are still some ambiguous or arbitrary steps or parameters in the study, I recommend that this paper only be accepted after the following major comments/issues are resolved:

1. The two represented phenotypes(starvation sensitive and sterile) were nicely investigated and presented. But in the current workflow, there are few steps that involve manual curation, like the selection of expression data and the representative GO terms. To show the application's universality, please add at least one additional test phenotype of interest that the related experiments and significant enrichment of GO terms were selected by predefined rules or algorithms.

2. The current flow diagram in Fig.1 is quite confusing, at least to me. Please consider separating the signature finding and phenotype prediction into two parts.

3. Meanwhile, since the paper already discussed the complexities of the relation between the gene-expression changes and the phenotypes and described the limitation of the current approach. If possible, I would like to see an evaluation part by the end of the signature-finding section to evaluate whether the selected phenotype could be significantly presented or identified by the expression profiles of the currently available microarray experiments.

4. In the cross-validation section, the paper shows "the AUC was calculated using the class (control/mutant) probabilities derived from the randomForest package, using the top 200 genes from the molecular signature (based on the p-values from the logistic regression)". How this 200 was determined, whether this top-N influence the AUCs? I would like to know the results of a series of Ns to have a better understanding of this step. And prefer a more rigorous or data-driven way to calculate or determine the numbers of the signature genes for different phenotypes.

Response:

We thank the reviewer for the helpful and thorough comments and suggestions. We are pleased to see that the Reviewer would like to see this approach further explored, tested, and improved by more computational researchers in the future and hopefully once the study is in the public domain, this would happen.

We have addressed all of the comments below:

1. The reviewer suggested to add one additional phenotype to test the general applicability of the methods. While we completely agree that adding more phenotypes will improve the overall methods and design, this will take considerable amount of time and effort. The overall goal of the presented work was to see if it was possible to implement a set of methods (combination of linear mixed effect models, GO terms and principal components) that could be used to predict complex phenotypes using gene-expression data. We also agree that there was scope for automating the generation of the molecular signatures, although this would require large amount of time and effort and was beyond the scope of our original hypothesis, i.e. is it possible to predict complex phenotypes using gene-expression data in Drosophila. 

This study is a proof-of-concept and we make this clear throughout the manuscript and this is why we feel that adding an additional phenotype is beyond the scope of this manuscript. 

2. We agree that the Figure 1 was confusing and we have separated the signature finding and phenotype prediction into two parts to make it clearer.

3. Yes, to make the evaluation part clearer, we have amended the discussion section to include a more detailed discussion if a particular phenotype can be represented or identified/predicted. We further described that it is possible that there could be phenotypes not well predicted by changes in gene-expression and these could for example be better represented or predicted using other types of data, for example methylation. In addition, the type of leave-one-out cross validation that we perform, tests precisely the question raised by the reviewer. That is, we leave one whole experiment, for example all the controls/mutants part of the crol experiment, create the molecular signature with the rest of the controls/mutants and test if we can predict the controls/mutants that were left out. This ensures that the overall AUC reflects the ability of the selected microarray experiments to predict the phenotype of interest. Of course, in order to gain a better understanding of what type of phenotypes and how many can be predicted with gene-expression data we need to investigate a large number of such phenotype gene-expression combinations. All of the above is described in detail in the Discussion section as suggested by the Reviewer.

4. We agree with the reviewer that the selection of the 200 genes to predict and calculate AUC is relatively arbitrary and that is why we performed a series of additional experiments, which are included in the revised manuscript. We selected a range of top genes and performed the leave-one-out cross-validation for each of these for all principal components. This ranged from 50 to 3,000 genes (15 different numbers of top genes). Overall, these were 120 leave-one-out cross-validations and AUCs for the starvation-sensitive and sterile phenotypes respectively. These new results are summarised in detail in supplementary figures 6 and 7 (Figs S6 and S7). Furthermore, we have also amended the Materials and Methods and Results sections to describe these analyses (sections "Leave-one-out cross-validation" and "Determining the number of PCs for unwanted variation" in the Materials and methods and Results respectively). For the starvation-sensitive phenotype there was little difference when choosing the number of top genes, although there is a trend for higher number of top genes to deliver higher AUC. For the sterile phenotype the opposite trend was noted, fewer genes resulted in better AUC. This could potentially be caused by the size of the transcriptional network responsible for the phenotype, for example it has been previously reported that the starvation stress resistance involves transcriptional response of ~25% of the genome in Drosophila. Nevertheless, this is a speculation and further work in terms of a large number of phenotypes need to be examined to assess this and gain further understanding. All of the above was described in detail in the Results section and data summarised in the Supplementary materials.

---

## [Decision Letter · Decision Letter 1]

5 Oct 2020

A novel computational approach for predicting complex phenotypes in Drosophila (starvation-sensitive and sterile) by deriving their gene expression signatures from public data

PONE-D-20-03484R1

Dear Dr. Ivanov,

We’re pleased to inform you that your manuscript has been judged scientifically suitable for publication and will be formally accepted for publication once it meets all outstanding technical requirements.

Kind regards,

Xia Li, Ph.D.

Academic Editor

PLOS ONE

Additional Editor Comments (optional):

My final decision is also "Accept"

Reviewers' comments:

Reviewer's Responses to Questions

**Comments to the Author**

1. If the authors have adequately addressed your comments raised in a previous round of review and you feel that this manuscript is now acceptable for publication, you may indicate that here to bypass the “Comments to the Author” section, enter your conflict of interest statement in the “Confidential to Editor” section, and submit your "Accept" recommendation.

Reviewer #2: All comments have been addressed

Reviewer #3: All comments have been addressed

Reviewer #4: All comments have been addressed

2. Is the manuscript technically sound, and do the data support the conclusions?

Reviewer #2: Yes

Reviewer #3: Yes

Reviewer #4: Yes

3. Has the statistical analysis been performed appropriately and rigorously? 

Reviewer #2: N/A

Reviewer #3: Yes

Reviewer #4: Yes

4. Have the authors made all data underlying the findings in their manuscript fully available?

Reviewer #2: Yes

Reviewer #3: Yes

Reviewer #4: Yes

5. Is the manuscript presented in an intelligible fashion and written in standard English?

Reviewer #2: Yes

Reviewer #3: Yes

Reviewer #4: Yes

6. Review Comments to the Author

Reviewer #2: Satisfied that comments are addressed in the various sections, abstract, intro and discussion. Can better clarify on the environmental influence part.

Reviewer #3: The authors have clarified most of the questions I raised in my previous review. I am glad to see the updated manuscript with a more focused title to reflect the study and clearer descriptions of the pipeline and the evaluation part.

Reviewer #4: I have a suggestion if the authors want to consider. The bracket in the title can be avoided with something like this:

A novel computational approach for predicting complex phenotypes in starvation-sensitive and sterile Drosophila by deriving their gene expression signatures from public data

7. PLOS authors have the option to publish the peer review history of their article (what does this mean?). If published, this will include your full peer review and any attached files.

Reviewer #2: No

Reviewer #3: No

Reviewer #4: No

---

## [Editor Report · Acceptance letter]

12 Oct 2020

PONE-D-20-03484R1 

A novel computational approach for predicting complex phenotypes in *Drosophila* (starvation-sensitive and sterile) by deriving their gene expression signatures from public data 

Dear Dr. Ivanov:

I'm pleased to inform you that your manuscript has been deemed suitable for publication in PLOS ONE. Congratulations! Your manuscript is now with our production department. 

Kind regards, 

on behalf of

Prof. Xia Li 

Academic Editor

PLOS ONE